# MAP4K4/JNK Signaling Pathway Stimulates Proliferation and Suppresses Apoptosis of Human Spermatogonial Stem Cells and Lower Level of MAP4K4 Is Associated with Male Infertility

**DOI:** 10.3390/cells11233807

**Published:** 2022-11-28

**Authors:** Cailin Wan, Wei Chen, Yinghong Cui, Zuping He

**Affiliations:** The Key Laboratory of Model Animals and Stem Cell Biology in Hunan Province, Hunan Normal University School of Medicine, The Manufacture-Based Learning & Research Demonstration Center for Human Reproductive Health New Technology of Hunan Normal University, Changsha 410013, China

**Keywords:** MAP4K4, human spermatogonial stem cells, JNK, proliferation, apoptosis

## Abstract

Spermatogonial stem cells (SSCs) serve as a foundation for spermatogenesis and they are essential for male fertility. The fate of SSC is determined by genetic and epigenetic regulatory networks. Many molecules that regulate SSC fate determinations have been identified in mice. However, the molecules and signaling pathways underlying human SSCs remain largely unclear. In this study, we have demonstrated that MAP4K4 was predominantly expressed in human UCHL1-positive spermatogonia by double immunocytochemical staining. MAP4K4 knockdown inhibited proliferation of human SSCs and induced their apoptosis. Moreover, MAP4K4 silencing led to inhibition of JNK phosphorylation and MAP4K4 phosphorylation at Ser801. RNA sequencing indicated that *MAP4K4* affected the transcription of *SPARC*, *ADAM19*, *GPX7*, *GNG2*, and *COLA1*. Interestingly, the phenotype of inhibiting JNK phosphorylation by SP600125 was similar to MAP4K4 knockdown. Notably, MAP4K4 protein was lower in the testes of patients with non-obstructive azoospermia than those with normal spermatogenesis as shown by Western blots and immunohistochemistry. Considered together, our data implicate that MAP4K4/JNK signaling pathway mediates proliferation and apoptosis of human SSCs, which provides a novel insight into molecular mechanisms governing human spermatogenesis and might offer new targets for gene therapy of male infertility.

## 1. Introduction

SSCs play an essential role in initialing and maintaining spermatogenesis. SSCs can both self-renew and differentiate into spermatocytes and subsequently mature spermatozoa after two meiotic divisions. Abnormality of the proliferation and/or differentiation causes non-obstructive azoospermia (NOA). It has been reported that SSCs from humans and non-human primates are different from rodents [1]. In human and non-human primates, SSCs could be characterized as A_dark_ or A_pale_ spermatogonia based upon cellular nuclei staining. The A_pale_ spermatogonia nuclei are hematoxylin-stained and appear light and more evenly pigmented, whereas the A_dark_ spermatogonia nuclei are dark and exhibit a sparse zone of unstained nuclei [2,3]. In rodents, the morphology of A_undiff_ spermatogonia is heterogeneous and characterized by formation of intercellular bridges among A_single_ (A_s_, solitary cells), A_paired_ (A_pr_, chain of two connected cells), and A_aligned_ (A_al_, chains of 4, 8, or 16 cells) cells [4]. Several markers, including GFRA1, SALL4, KIT, LIN28, and PLZF, are conserved across species, suggesting their roles in the formation of the spermatogenic lineage [5]. Significant variations exist in SSCs between humans and other species in terms of cell types, biological processes, and molecular functions. As an example, ID4 is expressed in the non-differentiated spermatogonia of humans [6] and mice [7], but not in non-human primates. In addition, the reproductive phenotype is not completely similar between humans and rodents.

The niches are a highly specialized microenvironment that facilitates self-renewal and differentiation of SSCs [8]. The function evaluation of SSCs was first reported using a transplantation assay in 1994 [9]. Glial cell line-derived neurotrophic factor (GDNF) has been demonstrated to be essential for SSC self-renewal [10]. Culture of germ cells derived from neonatal testes with the medium containing GDNF, epidermal growth factor (EGF), fibroblast growth factor 2 (FGF2), and leukaemia inhibitory factor (LIF) for over two years, results in the proliferation of SSCs and the formation of grape-like clusters on mouse embryonic fibroblasts [11]. This culture method is a milestone for obtaining sufficient cells to explore molecular mechanisms regulating fate decisions of SSCs. Nevertheless, no culture system has been established for expansion and long-term culture of human SSCs, which has become the handicap for basic research and translational application. Notably, we have set up a human SSC line with an unlimited potential of proliferation and high safety, which could provide a large number of cells for unveiling the molecules and signaling pathways of mediating human SSC fate determinations. 

It has been shown that Oct4 [12], Etv5 [13], Lhx1 [13], Bcl6b [13], Nanos2 [14], Id4 [7], and Cdc42 [15] are required for mouse SSC self-renewal. The microenvironment or niche for spermatogenesis consists of somatic cells, including Sertoli cells, Leydig cells, and peritubular myoid cells [16]. Sertoli cells protect and support male germ cells, whereas peritubular myoid cells maintain the integrity of testicular vasculature [17,18]. Sertoli cells and myoid cells secrete GDNF via LH-driven testosterone synthesis of Leydig cells [19]. GDNF has been demonstrated to activate Akt and Src family kinase (SFK) signaling in SSCs [20]. By activating the *Etv5* and *Bcl-b* genes, the FGF2-dependent MAP2K1 signaling pathway promotes SSC self-renewal [21]. Magnetic activated cell sorting (MACS) was utilized to obtain human SSCs, and SCF and retonic acid (RA) has been found by us to be important for the differentiation of human SSCs into round spermatids [22]. However, the molecules and signaling pathways that control human SSCs remain largely unclear.

In the current study, we have demonstrated that MAP4K4, encoding a protein belonging to the threonine/serine protein kinase family, was dominantly expressed in human testis. Significantly, we found that MAP4K4 modulated human SSC DNA synthesis, proliferation, and apoptosis using immunohistochemistry, Western blotting, CCK-8 and TUNEL assays. The SP600125 was employed to inhibit JNK phosphorylation, and phenotypic alterations were checked through the CCK-8 assay and fluorescence activated cell sorting (FACS). Immunoprecipitation was utilized to determine the interaction between MAP4K4 and JNK. RNA-sequencing was performed to identify the targets of MAP4K4, while immunohistochemistry and Western blotting were used to compare MAP4K4 levels between the testes of NOA and normal men. Our data showed that MAP4K4 was co-expressed with UCHL1 in human SSCs. MAP4K4 silencing reduced human SSC proliferation and enhanced their apoptosis. Notably, MAP4K4 stimulated JNK phosphorylation, whereas the inhibition of JNK phosphorylation by SP600125 resulted in the similar phenotypic changes as MAP4K4 knockdown. Moreover, RNA-sequencing showed that MAP4K4 controlled the transcriptions of multiple genes, e.g., *SPARC*, *ADAM19*, *GPX7*, *GNG2*, and *COLA1*. Significantly, the level of MAP4K4 protein was lower in the testis of NOA patients than in normal testis. These findings imply that MAP4K4 stimulates human SSC proliferation and suppresses their apoptosis via the MAP4K4/JNK signaling pathway and that abnormal levels of MAP4K4 might be associated with azoospermia. As such, this study offers a novel insight into the molecular mechanisms determining the fate decisions of human SSCs, and it might provide new targets for gene therapy for male infertility.

## 2. Materials and Methods

### 2.1. Collection and Treatment of Human Testis Tissues

This study was approved by The Ethical Review Committee of Hunan Normal University (ethics statement no.: 2018056), and the participants signed an informed consent. Testicular tissues were obtained from azoospermia patients at 25–65 years old and obstructive azoospermia (OA) patients with normal spermatogenesis at 35–39 years old. Testis tissues were rinsed thrice in PBS with 1% streptomycin and penicillin, and they were fixed in 4% paraformaldehyde (PFA) overnight. Subsequently, tissue embedding and sectioning were performed.

### 2.2. Culture of a Human SSC Line 

Our laboratory developed a human germline stem cell line with unlimited proliferation and no tumor formation via the transfection of a lentivirus (Lenti-EF1α-SV40LargeT-IRES-eGFP) [23]. This cell line expressed numerous markers of human germ cells and SSCs, e.g., VASA, GPR125, THY1, PLZF, GFRA1, UCHL1, and RET. The human SSC line was cultured in DMEM/F12 (Gibco, Grand Island, NY, USA) with 1% streptomycin/penicillin (Gibco) and 10% fetal bovine serum (Gibco) at 34 ℃ in 5% CO_2_ incubator. Every 4 days, cells were passaged by 0.05% trypsin and 0.53 mM EEDTA (Invitrogen, CA, USA). 

### 2.3. The siRNA Transfection

For knockdown experiments, synthesis of siRNA sequences targeting human *MAP4K4* mRNA was conducted by Genepharma (Shanghai, China) (Appendix A), whereas siRNAs without targeting sequences were utilized as the negative controls (NC). The lipofectamine^TM^ 3000 transfection reagent (Invitrogen, L3000015) was used to transfect siRNAs (Genepharma, Shanghai) into cells pursuant to the instruction of the manufacturer. At 48 h post-transfection, cells were harvested to assess changes in gene and protein expression.

### 2.4. RNA Extraction, RT-PCR, and qPCR

Cells were treated with RNAiso plus reagent (Takara, Kusatsu, Japan) to extract RNA according to the protocol of the manufacturer. RNA purity was analyzed based on the ratio of A_260_/A_280_ using the Nanodrop instrument (Thermo Scientific, Waltham, MA, USA). Total RNA was transcribed into complementary DNA (cDNA) by Evo M-MLV one step RT-PCR Kit (Accurate Biology, Changsha, China) as described previously [24]. The PCR reaction conditions were as follows: 5 min at 95 ℃, denaturation for 30 s at 95 ℃, annealing for 45 s at 52–60 ℃, and elongation for 45 s at 72 ℃ for 32 cycles. RNA without reverse transcription (RT) served as the negative control (NC). Electrophoresis with 2% agarose gel was used to separate PCR products, and visualization was performed using ethidium bromide.

Using the real-time PCR system (Applied Biosystems, Foster City, CA, USA), RNA was mixed with Power SYBR Green PCR Master Mix (Applied Biosystems, USA) to carry out qRT-PCR reactions. PCR products were quantified using the comparative CT (threshold cycle) method. The CT values of chosen genes were normalized against U6 or *ACTB* expression, and primer sequences were shown in Appendix A.

### 2.5. Immunocytochemistry and Immunofluorescence

The human SSC line was washed thrice using PBS (Gibco), fixed in 4% PFA for 15 min, washed thrice in cold PBS, and followed by permeabilization for 10 min using 0.5% Titon X-100 (Sigma, MO, USA). After extensive washing using cold PBS, cells were blocked with 5% BSA for 1 h at room temperature and followed by incubation at 4 ℃ with primary antibodies overnight. The sources and dilutions of antibodies were presented in Appendix A. Cells were rinsed with PBS and treated with Alexa Fluor 594-labeled IgG or Alexa Fluor 488-labeled IgG secondary antibodies. DAPI staining was performed for cell nuclei, and fluorescence microscopy (Leica, Wetzlar, Germany) was utilized to obtain cell images.

### 2.6. Western Blots

Testicular tissues and the human SSC line were lysed on ice with RIPA lysis buffer (Thermo Scientific) for approximately 30 min, and cell lysates were centrifugated at 12,000× *g* (Thermo Scientific) to obtain proteins. Thirty micrograms of total protein were separated on SDS-PAGE gels (Bio-Rad, CA, USA), and Western blot assays were conducted in terms of the method as previously described [24]. The antibodies and dilution ratios for this assay are shown in Appendix A, whereas protein blots were visualized by chemiluminescence (Bio-Rad).

### 2.7. CCK-8 Assay 

After siRNA transfection, human SSCs were incubated with culture medium and then with the 100 μL/mL CCK-8 reagent for 3.5 h. The results were analyzed with a microplate reader (Thermo Scientific) at 450 nm.

### 2.8. EdU Incorporation Assay

Human SSC line was incubated in 96-well plates (5000 cells/well), and each well had DMEM/F12 medium with 50 μmol/L EdU (RiboBio, Guangzhou, China). Cells were incubated for 12 h, washed with DMEM, and fixed in PFA (40 g/L). After neutralization using 2 mg/mL glycine, cells were permeabilized with 5 mL/L Triton X-100 for 10 min at room temperature. Apollo staining EdU staining was then displayed with reaction buffer, while Hoechst 33,342 was employed to stain cellular nuclei. Fluorescent microscopy (Leica, Wetzlar, Germany) was used for imaging, and the percentages of EdU-positive cells were determined by counting ≥500 cells.

### 2.9. Flow Cytometric Analysis

To quantify apoptosis of the human SSC line after MAP4K4-siRNAs transfection for 48 h, cells were digested, collected, and washed twice using cold PBS. Then, 10^6^ cells were resuspended in Annexin V Binding Buffer (BioLegend, London, UK) and supplemented with 5 μL APC Annexin V and 10 μL PI solution. After 15 min of incubation at room temperature, cells were subjected to flow cytometry assays on a C6 system (BD Biosciences, NJ, USA).

### 2.10. TUNEL Assay

An In Situ Cell Death Assay Kit (Roche, Mannheim, Germany) was used to evaluate apoptosis in the human SSC line transfected with MAP4K4 siRNAs. Cells were treated with proteinase K (20 mg/mL) for 15 min, followed by incubation with dUTP-labeled/terminal deoxynucleotidyl for an hour in the dark with dUTP-labeled/TdT enzyme buffer. Cell nuclei were stained with DAPI. Cells without TdT enzyme but with PBS were used as NC, and at least 500 cells/sample were assessed by Leica fluorescence microscopy.

### 2.11. RNA Sequencing

The Trizol kit (Invitrogen) was employed to extract total RNA. After quality assessment, mRNA was enriched with oligo (dT) beads, while the Ribo-Zero^™^ Magnetic Kit (Epicentre, Wisconsin, WI, USA) was utilized to eliminate rRNA for enrichment of mRNA. Enriched mRNA was fragmented in a fragmentation buffer and transcribed into cDNA by operating random hexamers. Synthesis of the second strand DNA was performed using DNA polymerase I, buffer, dNTP, and RNase H. Thereafter, the QiaQuick PCR extraction kit (Qiagen, Venlo, The Netherlands) was used to purify the cDNA fragments. The end repair, introduction of poly (A), and ligation by Illumina sequencing adapters were conducted. Agarose gel electrophoresis was carried out to elect ligation products, which was enriched by PCR and sequenced on an Illumina HiSeq2500 system. Sequencing was performed by Gene Denovo Biotechnology Company (Guangzhou, China). Reads were filtered out of the machine by fastp [25] (version 0.18.0), whereas rRNA mapped reads were deleted using the short reads alignment tool Bowtie2 (version 2.2.8) [26]. The remaining clean reads were utilized for assembly and gene abundance determination. To map the paired-end clean reads to the reference genome, the THISAT [27] was used. In a reference-based technique, the StringTie v1.3.1 [28] was utilized to construct the mapped reads. Differential expression levels of RNA were assessed using the DESeq2 [29], and *p* < 0.05 and absolute fold change ≥ 2 indicated the differentially expressed genes (DEGs) (Appendix A). DEGs in the human SSC line between MAP4K4-siRNA 2 and the control siRNA were evaluated by GO [30] and KEGG pathway analyses [31].

### 2.12. Identification of Cell Types and Analysis of Clusters

GSE109037 (11 samples) and GSE120508 (12 samples) were utilized to analyze scRNA-seq datasets using the Seurat software (R package, version 4.2; https://satijalab.org/seurat/). The Read.table or Read.csv function was used to first load the expression matrix data into R, and then Seurat objects were created from each test. Every assay used default parameters for filtering and normalization. Only cells that expressed over 500 genes and had less than 20% of reads match to the mitochondrial genome were preserved. The IntegrateData method was used to combine all the data after determining the variable properties of each item. After the mitochondrial and ribosomal genes were eliminated, the total datasets of the top 2500 DEGs were examined using UMAP and clustering. 

### 2.13. Statistical Analysis

All data were analysized with GraphPad Prism 8.0 (GraphPad Software, La Jolla, CA, USA) to detect the type of data distribution. The *t*-test was employed when the data were normally distributed and the variance was homogeneous, and the non-parametric test was applied for other data. Data were shown as mean ± SD from at least three independent experiments (*n* = 3). Groups were compared with the *t*-test, with *p* < 0.05 indicating significance.

## 3. Results

### 3.1. MAP4K4 was Highly Expressed in Human Spermatogonial Stem Cells

The MAP4K4 expression in adult human testes was determined through double immunohistochemical staining. We revealed that MAP4K4 was expressed in the cytoplasm of spermatogonia along the basement membranes of seminiferous tubules in normal adult human testes (Figure 1A), and about 83.94% of MAP4K4-expressing cells were UCHL1-positive spermatogonia (Figure 1C). We analyzed scRNA-seq datasets GSE109037 (11 samples) and GSE120508 (12 samples) of human adult testes with normal spermatogenesis in order to better understand human SSCs. Applying “Seurat” in R to minimize data dimensionality, the cells were separated into 14 populations (Appendix A). UMAP and clustering analyses of combined single-cell transcriptomic data from human testes were studied. Violin plots showed that MAP4K4 was highly expressed in spermatogonia, especially SSCs. Additionally, proliferating cell nuclear antigen (PCNA) was detectable in about 86.44% of MAP4K4-expressing cells (Figure 1B,D), indicating that MAP4K4 may contribute to cell proliferation in adult human testes. Together, these results suggest that MAP4K4 is predominantly expressed in proliferating human SSCs.

### 3.2. MAP4K4 Knockdown Inhibited Proliferation of Human SSCs

To explore the roles and mechanisms of MAP4K4 in mediating the fate decisions of SSCs, the human SSC line with biological features (Appendix A) of primary human SSCs were employed. To determine the influence of MAP4K4 knockdown on SSCs, siRNAs were employed to inhibit MAP4K4. Both MAP4K4-siRNA1 and MAP4K4-siRNA2 led to decreases in MAP4K4 mRNA and protein levels (Figure 2A–C), while MAP4K4-siRNA2 assumed more silencing effect of MAP4K4 (Figure 2A–C). A CCK8 assay showed that MAP4K4-siRNA2 suppressed proliferation of human SSC line on days 3 to 5 following siRNA transfection (Figure 2D). Western blots displayed that the expression level of PCNA, a marker for cell proliferation, was decreased in human SSC lines treated with MAP4K4-siRNA2 (Figure 2E,F). After 48 h of transfection with MAP4K4-siRNA2, the percentage of EdU-positive cells was significantly lower than the control siRNA. (30.48% ± 2.05 vs. 17.51% ± 0.43, *p* < 0.05) (Figure 2G–I). Taken together, these findings implicate that MAP4K4 knockdown inhibits human SSC proliferation and DNA synthesis.

### 3.3. MAP4K4 knockdown Enhanced Apoptosis of Human SSCs

Since MAP4K4 knockdown increased cell debris in cultured cells, the significance of MAP4K4 in the regulation of human SSC apoptosis was investigated utilizing annexin V/propidium iodide (PI) labeling and flow cytometry. Compared to the control siRNA, the early and late apoptotic rates of human SSC line were both significantly enhanced by MAP4K4 knockdown (Figure 3A–C). A TUNEL assay showed that MAP4K4-siRNA2 increased the percentages of TUNEL-positive cells when compared with the control siRNA (12.34% ± 2.10 vs. 23.18% ± 2.12, *p* < 0.05) (Figure 3D–F). Considered together, these data indicate that MAP4K4 knockdown enhances the apoptosis of human SSCs.

### 3.4. MAP4K4 Interacted with JNKs and Mediated JNK Phosphorylation in the Human SSC line

It has been reported that JNK phosphorylation is mediated by MAP4K4 [32,33]. To elucidate whether MAP4K4 interacts with JNKs, immunoprecipitation (IP) and Western blots were performed. Our Co-IP assay indicated that MAP4K4 could bind to JNKs (Figure 4A) and that JNKs were able to pull down MAP4K4 (Figure 4B). Expression levels of phosphorylated JNKs were detected using Western blots in the human SSC line after transfection with MAP4K4-siRNA2. We found that the phosphorylated JNKs levels were significantly decreased by MAP4K4-siRNA2 (Figure 4C,D). It has been shown that MAP4K4 phosphorylation at Ser801 is a highly conserved location [34], and we revealed that the level of MAP4K4 phosphorylation at Ser801 was decreased in the human SSC line after MAP4K4 knockdown (Figure 4E,F).

### 3.5. JNK Phosphorylation Inhibition Affected Proliferation and Apoptosis of the Human SSC Line

We next explored whether MAP4K4 regulated the proliferation and apoptosis of the human SSC line via JNK phosphorylation. To test this hypothesis, we blocked the JNK activity with a selective JNK phosphorylation inhibitor SP600125. After 24 h of exposure to SP600125, the level of phosphorylated JNKs in the human SSC line was lower than the control group (Figure 5A,B). No change in the JNKs protein was observed in the human SSC line between SP600125 treatment or the control (Figure 5C). Subsequently, the CCK-8 assay was performed to assess the proliferation of human SSCs. Treatment with SP600125 for 3–5 days significantly decreased proliferation of the human SSC line (Figure 5D). Moreover, the percentage of EdU-positive cells was reduced in the human SSC line following 24 h of treatment with SP600125 (Figure 5E–G) [25.23 ± 0.24 (control) vs. 18.56 ± 0.37 (SP600125), *p* < 0.05]. Annexin V/propidium iodide staining and flow cytometry revealed that SP600125 increased apoptosis of the human SSC line [10.33% ± 1.24% (control) vs. 16.00% ± 1.41% (SP600125), *p* < 0.05] (Figure 5H–G). Collectively, these data suggest that inhibition of JNK phosphorylation through SP600125 decreased the proliferation of human SSCs and enhanced their apoptosis.

### 3.6. Screening of MAP4K4 Target Genes in Human SSCs

To identify the target genes of MAP4K4 in human SSCs, RNA sequencing of the human SSC line subjected to MAP4K4-siRNA2 or control-siRNA was performed. A total of 20,289 genes were sequenced. DEGs were selected based upon *p* < 0.05 and fold change ≥ 2 thresholds. The heat map and volcano plot shown in Figure 6A,B demonstrate that 37 genes were unregulated and 37 genes were down-regulated by MAP4K4-siRNA2. Five genes, including *SPARC*, *ADAM19*, *GNG2*, *GPX7*, and *COLA1*, were randomly selected to verify the RNA sequencing data through RT-PCR (Figure 6C). GO analysis showed that MAP4K4 affected biological processes, including cellular processes, metabolic processes, and biological regulation, as well as regulating molecular functions, e.g., binding, catalytic activity, and transcription regulator activity. In addition, KEGG analysis revealed significant differences in protein digestion and absorption and relaxin signaling pathways after MAP4K4 silencing. Among the top ten downregulated genes, *SPARC* and *GNG2* mediated biological processes and pathways associated with cell proliferation and apoptosis, respectively (Figure 6D,E).

### 3.7. Decreased Expression Level of MAP4K4 in NOA

NOA is one of the most severe conditions of male infertility, and it is characterized by the absence of sperm in the ejaculate owing to spermatogenesis failure. NOA can be classified as follows: spermatogonial arrest, spermatocyte arrest, spermatid maturation arrest, hypospermatogenesis, and Sertoli cell only syndrome. Male infertility is a heterogeneous, environmental, and genetically determined reproductive system disorder that affects 1% of men and 10%–20% of infertile males [35,36]. When both distal and proximal ejaculatory ducts are blocked bilaterally, normal spermatogenesis (Appendix A) is unaffected, which leads to obstructive azoospermia (OA). NOA is caused by the failure of primary or secondary testicular spermatogenesis. MAP4K4 expression was evaluated histopathologically in the testis tissues of five patients to determine whether it is associated with spermatogenesis failure (Appendix A). Western blots demonstrated that MAP4K4 protein levels were remarkably lower in human testis tissues of NOA patients compared with OA patients with normal spermatogenesis (Figure 7A,B). Compared to OA patients with normal spermatogenesis, the proportion of MAP4K4-expressing cells in UCHL1-positive SSCs was reduced in NOA (Figure 7C,D), as shown by double immunohistochemistry. Taken together, these findings indicate that lower levels of MAP4K4 expression might be associated with abnormal spermatogenesis.

## 4. Discussion

Studies on the genes and signaling pathways regulating human SSCs are relatively fewer compared to rodents due to a variety of reasons. The resources of human testis and SSCs for research is affected by legal and ethical issues, and it is difficult to culture and expand human primary SSCs. The human SSC line we established was used in this study because of its potential to proliferate indefinitely [23,37].

Based upon data from single-cell sequencing datasets GSE109037 (11 samples) and GSE120508 (12 samples), we found that MAP4K4 was primarily expressed in proliferating SSCs (UCHL1^+^/PCNA^+^). MAP4K4 is engaged in the regulation of stemness maintenance and differentiation in many stem cells. For examples, MAP4K4 promotes the proliferation of retinal stem cells [38], inhibits the differentiation of adipose stem cells, and suppresses adipogenesis [33,39]. In cancer cells, downregulation of MAP4K4 increases apoptosis and cell cycle arrest and inhibits cell proliferation [40,41,42]. It has been shown that MAP4K4 inhibits the survival of cardiomyocytes and motor neurons derived from human pluripotent stem cells [43,44]. Recently, MAP4K4 has been shown to play a critical role in LATS-YAP regulation in mouse embryonic fibroblasts [45]. RAS-associated RAP2 binds to and stimulates several kinases, including MAP4K4, which leads to activation of LATS2 and LATS1 as well as the suppression of TAZ and YAP [46]. These studies demonstrate that MAP4K4 acts as an upstream regulator of critical transduction effectors in cell proliferation regulation. Inhibitors of MAP4K4 are currently being developed in clinical trials as a candidate treatment for neurodegenerative diseases by increasing the activity of motor neurons extracted from the induced pluripotent stem cells (iPSCs) [47]. It has been reported that MAP4K4 plays an important role in the development of mammals. *MAP4K4*
^(-/-)^ mouse embryos fail to develop somites and die during the embryonic days (E) 9.5 and E10.5 [48]. Additionally, prenatal exome sequencing of an abnormal fetus shows a de novo nonsense mutation in *MAP4K4,* which may represent a new candidate for a human developmental disease [49]. MAP4K4 is an upstream activator of the JNK signaling pathway [50,51]. JNK is one of the potential regulatory candidates involved in the mitotic-to-meiotic transition prior to early preleptotene spermatocytes [52]. Significantly, ROS is a downstream effector of Ras signaling that modulates the phosphorylation of JNKs to control the proliferation of mouse SSCs [53]. We were eager to clarify the molecular mechanisms in which they modulate the human SSCs. Here, we have demonstrated that MAP4K4 silencing inhibited proliferation of human SSCs and induced their apoptosis. Notably, MAP4K4 knockdown resulted in inhibition of JNK phosphorylation and MAP4K4 phosphorylation at Ser801.

Our RNA sequencing and differential gene enrichment analysis indicate that MAP4K4 targets genes were involved in collagen formation and extracellular matrix organization. Moreover, the MAP4K4 protein affected the transcription of important genes, including *SPARC*, *ADAM19*, *GPX7*, *GNG2*, and *COLA1*. However, it remains unknown whether MAP4K4 influences these genes via JNK, and further research of the signaling pathway is required. Interestingly, SPARC has been shown to be associated with gonadal development and spermatogenesis in mammals [54,55,56]. Moreover, SPARC has been shown to have a function in ocular surface healing, which is mediated by the phosphorylation of JNK and p38-MAPK signaling pathways [57]. It has been suggested that MAP4K4 affects SH3PXD2A-binding enzymes ADAM12, ADAM15, and ADAM19, which has been shown to be associated with infertility in mutant mouse models [58]. Furthermore, ADAM19 modulates sperm-egg interaction and fusion [59,60]. According to KEGG analysis, MAP4K4 is associated with the PI3K-Akt signaling pathway, which is consistent with previous findings that the MAPK signal pathway regulates mammalian cell proliferation [61]. Recent studies [62,63,64] have found that the MAPK and PI3K pathways interact, and PI3-kinase activity contributes to epidermal growth factor (EGF)-induced JNK activation [65]. Sertoli cells, Leydig cells, peritubular myoid cells, and other testicular somatic cells secrete cytokines and growth factors, e.g., GDNF [8], FGF2 [12], EGF [66], and LIF [11], which are essential for the proliferation and differentiation of SSCs. The findings mentioned above suggest that MAP4K4 may participate in cell-to-cell communication between somatic cells and stem cells.

The scRNA-seq analysis revealed that MAP4K4 was also expressed at a lesser level in differentiated male germ cells, including differentiating spermatogonia. Therefore, it remains unclear whether MAP4K4 promotes or inhibits cell differentiation of SSCs. During spermatogenesis, the expression level of MAP4K4 is decreased gradually. It has been reported that MAP4K4 acts as an activator of the JNK signaling pathway and regulates brown adipose tissue development [67]. Additionally, distinct MAP4K4 isoforms have different roles in adipogenesis. For examples, MAP4K4 isoform 1 represses brown adipogenesis, whereas MAP4K4 isoform 4 improves brown adipocyte physiological characteristics [67]. As such, we hypothesize that MAP4K4 inhibits the differentiation of spermatogonia, whereas the existence of MAP4K4 variants with alternative splicing may play distinct roles.

We found that the level of MAP4K4 was significantly lower in NOA patients, reflecting that aberrant expression of MAP4K4 may be associated with NOA occurrence. However, more samples are needed to verify this data. Likewise, screening for mutations of MAP4K4 in NOA patients would be performed in future studies and mouse knockout models can be utilized to investigate whether MAP4K4 deletion leads to dysregulation of spermatogenesis.

## 5. Conclusions

In this study, we found MAP4K4 was highly expressed in human SSCs and that MAP4K4 regulated SSC proliferation and apoptosis through MAP4K4 phosphorylation and JNK phosphorylation. Furthermore, we found that the level of MAP4K4 was significantly lower in NOA, reflecting that MAP4K4 may be associated with the development of NOA. Collectively, this study provides novel insights into the molecular mechanisms underlying SSC fate determinations and offers new targets for male infertility treatment.

## Figures and Tables

**Figure 1 cells-11-03807-f001:**
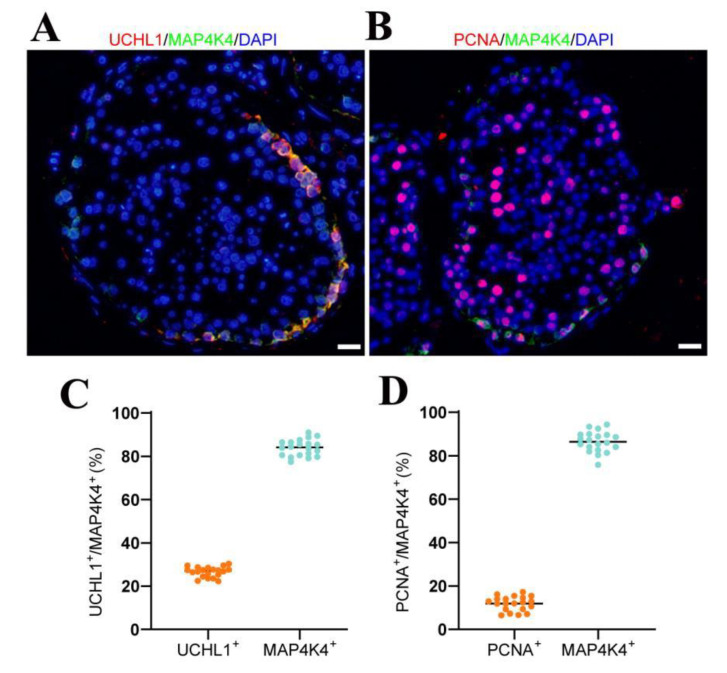
The expression of MAP4K4 in normal human testis. (**A**,**B**) Double immunostaining showed the co-expression of MAP4K4 with UCHL1 (**A**) and PCNA (**B**) in normal human testis. (**C**,**D**) The percentages of MAP4K4^+^ cells with UCHL1 (**C**) and PCNA (**D**) expression. Each dot in the graph represented the positive cells in one seminiferous tubule, and black line was the mean value. At least 20 seminiferous tubules were counted. Scale bars in A: 50 μm.

**Figure 2 cells-11-03807-f002:**
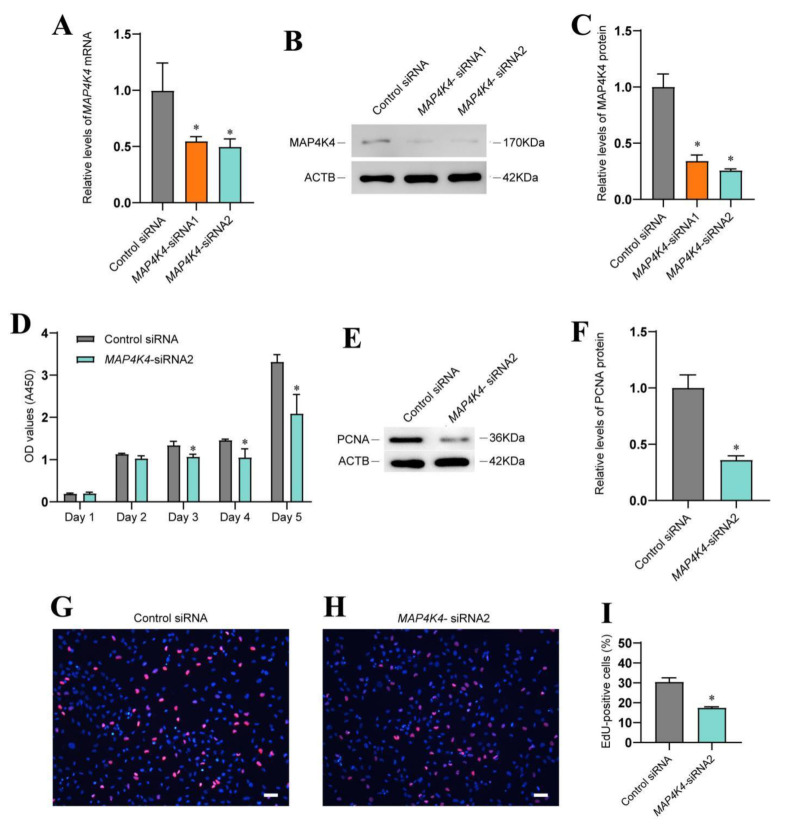
The influence of MAP4K4 knockdown on human SSC proliferation. (**A**) QPCR revealed MAP4K4 mRNA levels in a human SSC line after treatment with MAP4K4-siRNA1 and 2. (**B**,**C**) Western blotting showed MAP4K4 protein level changes in the human SSC line after the transfection of MAP4K4-siRNA1 and 2. ACTB was used as the loading control for the total protein. (**D**) CCK-8 assay illustrated the proliferation of human SSC line transfected with the control siRNA and MAP4K4-siRNA2. (**E**,**F**) The relative levels of PCNA protein in human SSC line after transfection with the control siRNA and MAP4K4-siRNA2. (**G**–**I**) The percentages of EdU-positive cells in human SSC line transfected with control siRNA and MAP4K4-siRNA2. Scale bars in G: 50 μm. * denote *p* < 0.05.

**Figure 3 cells-11-03807-f003:**
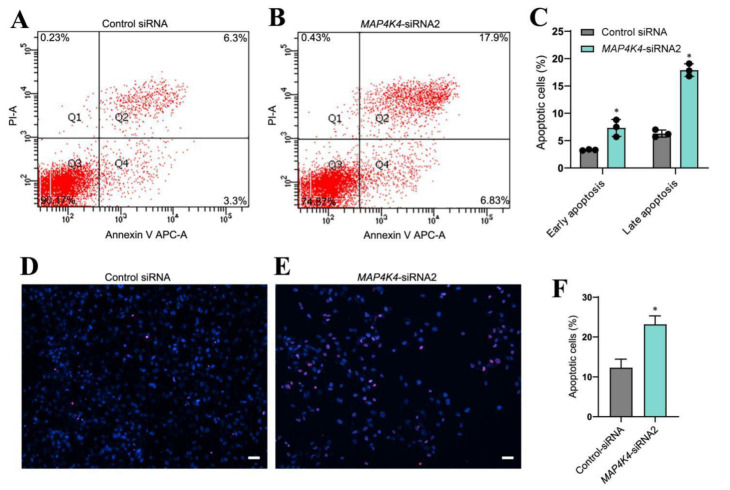
The effect of MAP4K4 knockdown on the apoptosis of the human SSC line. (**A**–**C**) Flow cytometric and APC Annexin V analysis showed the percentages of apoptosis in human SSC line after transfection with control siRNA and MAP4K4-siRNA2. (**D**–**F**) TUNEL assays displayed the percentages of TUNEL^+^ cells in human SSC line with transfection of control siRNA and MAP4K4-siRNA2. Scale bars in C: 50 μm. * indicate *p* < 0.05.

**Figure 4 cells-11-03807-f004:**
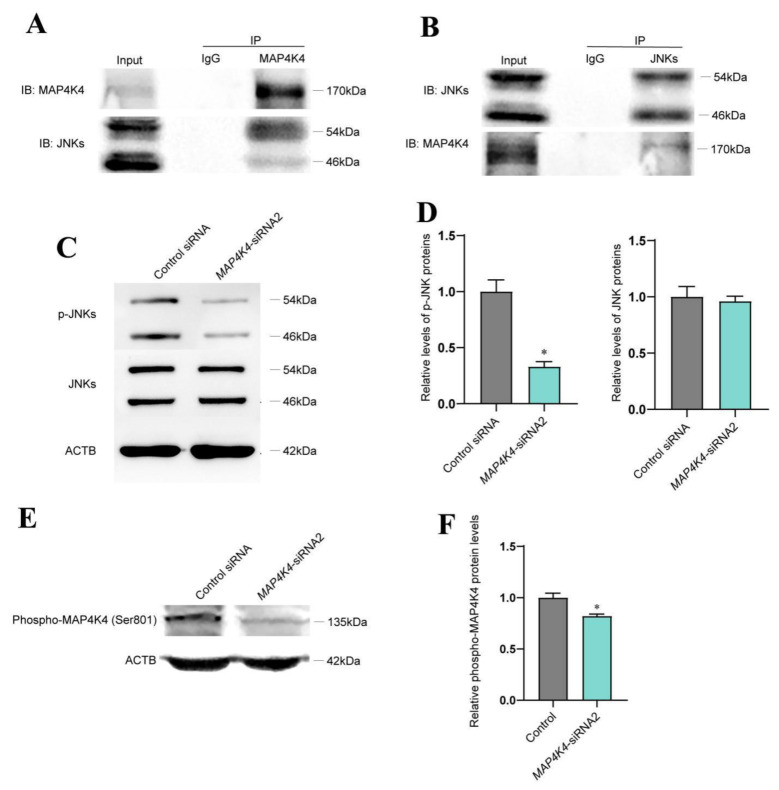
MAP4K4 silencing reduced the phosphorylation of JNK in human SSC line. (**A**,**B**) The Co-IP assay indicated an interaction between MAP4K4 and JNK. (**C**,**D**) Western blotting showed the level changes of p-JNK protein in human SSC line after MAP4K4 knockdown. (**E**,**F**) Western blotting showed that at 72 h after transfection with MAP4K4-siRNA2, the ratio of p-MAP4K4 at Ser801 to total MAP4K4 in human SSC line was decreased. * represents *p* < 0.05.

**Figure 5 cells-11-03807-f005:**
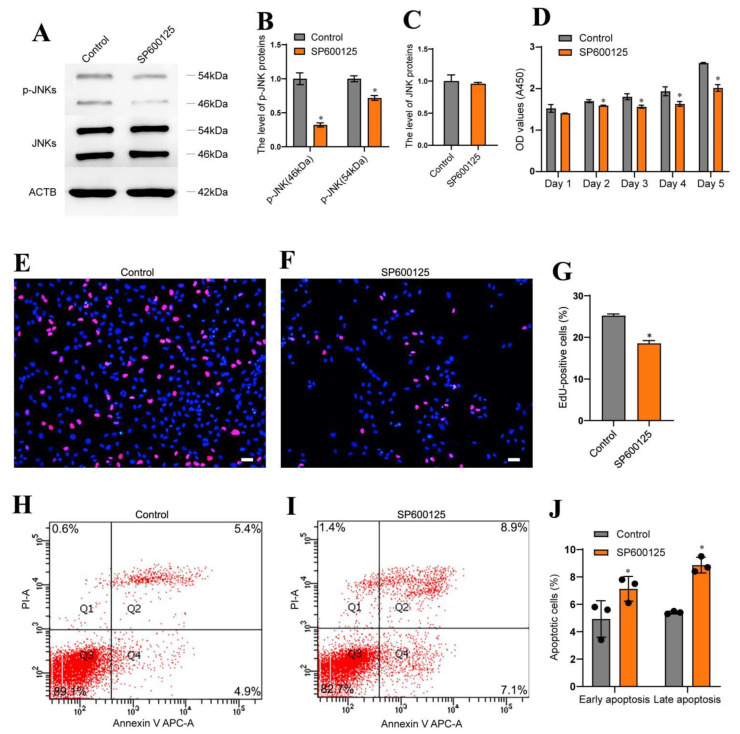
Inhibition of JNK phosphorylation decreased human SSC proliferation and enhanced their apoptosis. (**A**–**C**) Western blotting showed that the phosphorylation of JNKs was significantly decreased in human SSC line by the JNKs phosphorylation inhibitor SP600125. (**D**) CCK-8 assay displayed the proliferation of human SSC line after treat with SP600125 from day 1 to day 5. (**E**–**G**) EdU incorporation assay revealed DNA synthesis in human SSC line after treatment with SP600125 for 48 h. Scale bars in D: 50 μm. (**H**–**J**) Flow cytometry and APC/Annexin V analysis showed the proportions of apoptosis in human SSC line treated with SP600125 for 48 h. * indicates *p* < 0.05.

**Figure 6 cells-11-03807-f006:**
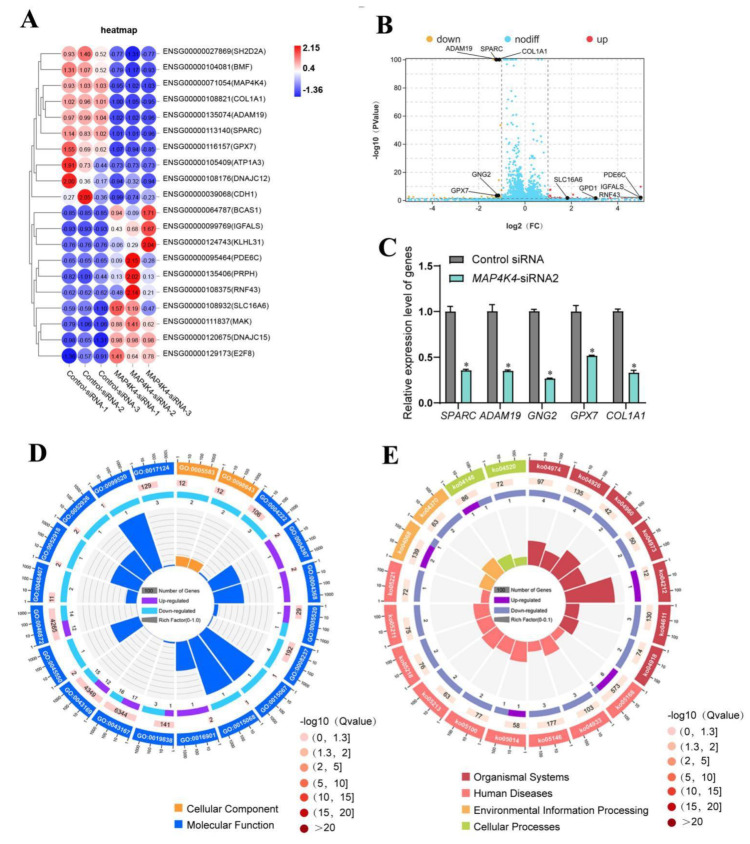
The identification of MAP4K4 target genes in human SSC line. (**A**) Hierarchical clustering demonstrates the top 20 DEGs in the human SSC line between control siRNA and MAP4K4-siRNA2. (**B**) Volcano plots show the differences between control siRNA and MAP4K4-siRNA2. (**C**) QPCR evaluated the mRNA in *SPARC*, *ADAM19*, *GNG2*, *GPX7*, and *COLA1* in human SSC line by control siRNA and MAP4K4-siRNA2. (**D**) GO circle plot illustrated the top 20 enrichment terms. First circle: GO terms of the top 20 enrichments, and outside circle was the sitting scale of the number of genes. Different colors denoted the different ontologies; the second circle: the number of DEGs was enriched in this GO term and the *Q*-value. The more genes, the longer the bar; the smaller the *Q*-value, the more red the color; the third circle: the bar chart of the proportion of DEGs, and purple denoted the fraction of up-regulated genes. Light blue denoted the proportion of down-regulated genes, and the specific value was indicated below; the fourth circle: Rich Factor value of each GO term (the number of differences in this GO term was divided by all the numbers), background grid line, each grid indicates 0.1. (**E**) KEGG circle plot for top 20 enrichment pathways. The first circle: top 20 enriched pathways, outside circle was the coordinate scale of gene number, and different colors signified different classes; the second circle: the number of DEGs enriched in this GO term and the *Q*-value. The more genes, the longer the bar; the smaller the *Q*-value, the more red the color; the third circle: the bar chart of the proportion of DEGs; purple indicated the fraction of up-regulated genes, while blue represented the proportion of down-regulated genes; the specific value was indicated below; the fourth circle: Rich Factor value of each pathway (the number of differences in this pathway was divided by all the numbers), background grid line, each grid denotes 0.1. * represented *p*< 0.05.

**Figure 7 cells-11-03807-f007:**
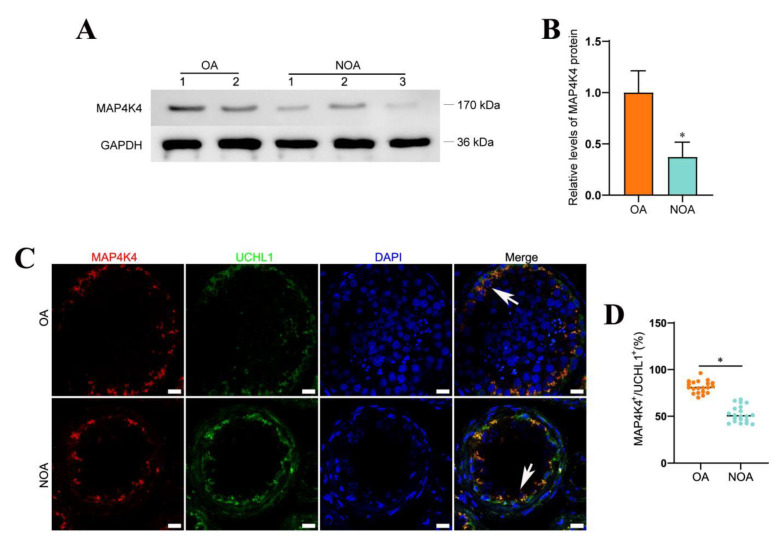
The expression of MAP4K4 in OA and NOA patients. (**A**,**B**) Western blotting showed the levels of MAP4K4 protein between OA and NOA patients. (**C**,**D**) The percentages of UCHL-positive cells expressing MAP4K4 (arrows) in OA and NOA patients. Scale bars in C: 100 μm. * indicates *p* < 0.05.

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
