# Peer review of "MAP4K4/JNK Signaling Pathway Stimulates Proliferation and Suppresses Apoptosis of Human Spermatogonial Stem Cells and Lower Level of MAP4K4 Is Associated with Male Infertility"

_cells, 2022, doi:10.3390/cells11233807_

Round 1
Reviewer 1 Report
The authors examined expression and functional roles of MAP4K4 kinase in human spermatogonial stem cells. Using both in vitro cultured hSSC cell lines and patient testis samples, they found that MAP4K4 is highly enriched in UCHL1-expressing hSSCs, whereas knockdown of Map4k4 expression via siRNA method caused reduced proliferation and increased apoptosis in cultured hSSCs. In addition, the author found through co-IP experiment, that MAP4K4 complexed with JNK, and affected JNK phosphorylation when being knocked down. Using a small molecule inhibitor of JNK, they found that inhibition of JNK exhibited similar proliferation and apoptotic phenotypes as the knock-down of Map4k4 in cultured hSSCs. These data provided evidences of potential roles of MAP4K4 during hSSC regulation and an interesting link between MAP4K4 and JNK signaling. The results of reduced MAP4K4 in NOA patients’ testis is also interesting for potential etiological studies.
The results in large part support the conclusions of enriched expression of MAP4K4 in hSSCs, and the participation of MAP4K4 in regulating the proliferation of hSSCs. The phenotypic analyses in cell culture also in agreement with the link between MAP4K4 and JNK during the proliferation of hSSCs. However, the manuscript contains overstatements on some of the data that should be modified and interpreted more carefully.
1. The authors found that MAP4K4 silencing reduced phosphorylation of S801. This is done by siRNA method in which MAP4K4 protein was reduced. It did not show whether S801 phosphorylation would be a critical site for mediating MAP4K4 functions in hSSCs.
2. Co-immunoprecipitation experiments help to find protein complexes, however, not necessarily indicating direct interactions between two proteins;
3. It is not clear whether the primary function of MAP4K4 is to regulate cell proliferation or to prevent cells from apoptosis. In Figure 3B, what is the time range for early and late apoptosis? Did the authors examine the cell proliferation by the time frame?
4. From the analyses of human samples, it is not clear how the reduced MAP4K4 expression conveyed in these NOA patients, could the authors provide some clues in the manuscript.
Some other concerns:
1. The Summary may be rephrased for: “MAP4K4 silencing …. MAP4K4 phosphorylation at Ser801”, “MAP4K4 affected transcription of ….”, “MAP4K4/JNK pathway mediates proliferation and apoptosis .… ”.
2. Provide the sample size and experimental repeats in all figure legends, this can be done by indicating N=?
3. Provide references for GSE datasets used.
4. Can the authors also provide transmission light images of hSSCs in Figure 2G, 3C and 5D.
5. The font in Figure 6 may be enlarged for readability.
6. Check the text for spelling errors.
Author Response
Response to Reviewer 1 comments
The authors examined expression and functional roles of MAP4K4 kinase in human spermatogonial stem cells. Using both in vitro cultured hSSC cell lines and patient testis samples, they found that MAP4K4 is highly enriched in UCHL1-expressing hSSCs, whereas knockdown of Map4k4 expression via siRNA method caused reduced proliferation and increased apoptosis in cultured hSSCs. In addition, the author found through co-IP experiment, that MAP4K4 complexed with JNK, and affected JNK phosphorylation when being knocked down. Using a small molecule inhibitor of JNK, they found that inhibition of JNK exhibited similar proliferation and apoptotic phenotypes as the knock-down of Map4k4 in cultured hSSCs. These data provided evidences of potential roles of MAP4K4 during hSSC regulation and an interesting link between MAP4K4 and JNK signaling. The results of reduced MAP4K4 in NOA patients’ testis is also interesting for potential etiological studies.
We thank the reviewer 1 for the nice comments that our study is interesting.
The results in large part support the conclusions of enriched expression of MAP4K4 in hSSCs, and the participation of MAP4K4 in regulating the proliferation of hSSCs. The phenotypic analyses in cell culture also in agreement with the link between MAP4K4 and JNK during the proliferation of hSSCs. However, the manuscript contains overstatements on some of the data that should be modified and interpreted more carefully.
We have carefully revised our manuscript in terms of the helpful comments and suggestions of this reviewer.
- The authors found that MAP4K4 silencing reduced phosphorylation of S801. This is done by siRNA method in which MAP4K4 protein was reduced. It did not show whether S801 phosphorylation would be a critical site for mediating MAP4K4 functions in hSSCs.
We have now removed the sentence at lines 289-290.
- Co-immunoprecipitation experiments help to find protein complexes, however, not necessarily indicating direct interactions between two proteins;
We have deleted “direct” on the legend of Figure 4, page 9, line 292.
- It is not clear whether the primary function of MAP4K4 is to regulate cell proliferation or to prevent cells from apoptosis. In Figure 3B, what is the time range for early and late apoptosis? Did the authors examine the cell proliferation by the time frame?
We performed the functional assays demonstrating that MAP4K4 regulates cell proliferation and apoptosis of human SSCs as shown in Figures 2 and 3.
- From the analyses of human samples, it is not clear how the reduced MAP4K4 expression conveyed in these NOA patients, could the authors provide some clues in the manuscript.
Our Western blots demonstrated that MAP4K4 protein levels were remarkably lower in human testis tissues of NOA patients compared with OA patients with normal spermatogenesis (Figure 7A, 7B). Compared to OA patients with normal spermatogenesis, the proportion of MAP4K4-expressing cells in UCHL1-positive SSCs was reduced in NOA (Figure 7C, 7D) as shown by double immunohistochemistry. Taken together, these findings indicate that lower levels of MAP4K4 expression might be associated with abnormal spermatogenesis.
Some other concerns:
- The Summary may be rephrased for: “MAP4K4 silencing …. MAP4K4 phosphorylation at Ser801”, “MAP4K4 affected transcription of ….”, “MAP4K4/JNK pathway mediates proliferation and apoptosis .… ”.
Our summary has been shown as follows: In this study, we found MAP4K4 was highly expressed in human SSCs and that MAP4K4 regulated SSC proliferation and apoptosis through MAP4K4 phosphorylation and JNK phosphorylation. Furthermore, we found that the level of MAP4K4 was significantly lower in NOA, reflecting that MAP4K4 may be associated with the development of NOA.
- Provide the sample size and experimental repeats in all figure legends, this can be done by indicating N=?
Lines 218, we have now shown that ‘Data were shown as mean ± SD from at least three independent experiments (n=3)’.
- Provide references for GSE datasets used.
We have now shown the datasets GSE109037 (11 samples) and GSE120508 (12 samples) used in this study.
- Can the authors also provide transmission light images of hSSCs in Figure 2G, 3C and 5D.
We have included DAPI staining to show the cell nuclei of human SSCs in Figure 2G, 3C and 5D.
- The font in Figure 6 may be enlarged for readability.
We have now enlarged the Figure 6.
- Check the text for spelling errors.
We have carefully checked the text of the manuscript to ensure no grammar or spelling error.
Reviewer 2 Report
The manuscript "MAP4K4/JNK signaling pathway stimulates proliferation and suppresses apoptosis of human spermatogonial stem cells and lower level of MAP4K4 is associated with male infertility" the authors demonstrate in human SSC thet MAP4K4/ JNK pathway increase proliferation and decreases apoptosis and the MAP4K4 levels can be associated with infertility. The manuscript is well written and is a pleasure to read.
The authors addressed the proposed questions. The proposed methods are adequate to the study; the results are presented clearly; and in the discussion section the results obtained are supported properly. The topic is interesting and relevant to the research field, this study can help to underline possible biomarkers for male infertility in the near future. For me the manuscript is able to be accepted for publication.
Author Response
Response to Reviewer 2 comments
The manuscript "MAP4K4/JNK signaling pathway stimulates proliferation and suppresses apoptosis of human spermatogonial stem cells and lower level of MAP4K4 is associated with male infertility" the authors demonstrate in human SSC thet MAP4K4/ JNK pathway increase proliferation and decreases apoptosis and the MAP4K4 levels can be associated with infertility. The manuscript is well written and is a pleasure to read.
The authors addressed the proposed questions. The proposed methods are adequate to the study; the results are presented clearly; and in the discussion section the results obtained are supported properly. The topic is interesting and relevant to the research field, this study can help to underline possible biomarkers for male infertility in the near future. For me the manuscript is able to be accepted for publication.
We thank the reviewer 2 for the positive comments that our manuscript is well written and can be accepted and that our study is interesting.
Reviewer 3 Report
The manuscript is very interesting and it add an important part for deepen our understanding of the molecular mechanisms involved in the proliferation and differentiation of SSCs. The authors used comprehensive methodology to provide precise results.
This reviewer suggested adding arrows (at least in Fig. 7C) to show the stained cells. Also, it is important to show a co-staining of MAP4K4 with additional markers of SSCs such as SALL4 and/or PLZF in the examined testicular tissues. Do the UCHL1 cells that positively stained to MAP4K4 also stained to PCNA?
Author Response
Reviewer 3 comments
The manuscript is very interesting and it adds an important part for deepen our understanding of the molecular mechanisms involved in the proliferation and differentiation of SSCs. The authors used comprehensive methodology to provide precise results.
We feel grateful to the reviewer 3 for the nice comments that our study is very interesting.
This reviewer suggested adding arrows (at least in Fig. 7C) to show the stained cells. Also, it is important to show a co-staining of MAP4K4 with additional markers of SSCs such as SALL4 and/or PLZF in the examined testicular tissues. Do the UCHL1 cells that positively stained to MAP4K4 also stained to PCNA?
We have now added arrows to showing the co-expression of MAP4K4 and UCHL1.
We have also shown the co-expression of MAP4K4 and UCHL1 in Figure 1A and 1B as well as MAP4K4 and PCNA in Figure 1B and 1D.
Round 2
Reviewer 3 Report
None